# The Cross-Sectional Intrinsic Entropy—A Comprehensive Stock Market Volatility Estimator

**DOI:** 10.3390/e24050623

**Published:** 2022-04-29

**Authors:** Claudiu Vințe, Marcel Ausloos

**Affiliations:** 1Department of Economic Informatics and Cybernetics, Bucharest University of Economic Studies, 010552 Bucharest, Romania; 2School of Business, Brookfield, University of Leicester, Leicester LE2 1RQ, UK; ma683@le.ac.uk; 3Department of Statistics and Econometrics, Bucharest University of Economic Studies, 010374 Bucharest, Romania; 4GRAPES (Group of Researchers for Applications of Physics in Economy and Sociology), 483 Rue de la Belle Jardiniere, B-4031 Liege, Belgium

**Keywords:** intrinsic entropy, cross-sectional study, stock market, volatility estimator

## Abstract

To take into account the temporal dimension of uncertainty in stock markets, this paper introduces a cross-sectional estimation of stock market volatility based on the intrinsic entropy model. The proposed cross-sectional intrinsic entropy (*CSIE*) is defined and computed as a daily volatility estimate for the entire market, grounded on the daily traded prices—open, high, low, and close prices (OHLC)—along with the daily traded volume for all symbols listed on The New York Stock Exchange (NYSE) and The National Association of Securities Dealers Automated Quotations (NASDAQ). We perform a comparative analysis between the time series obtained from the *CSIE* and the historical volatility as provided by the estimators: close-to-close, Parkinson, Garman–Klass, Rogers–Satchell, Yang–Zhang, and intrinsic entropy (*IE*), defined and computed from historical OHLC daily prices of the Standard & Poor’s 500 index (S&P500), Dow Jones Industrial Average (DJIA), and the NASDAQ Composite index, respectively, for various time intervals. Our study uses an approximate 6000-day reference point, starting 1 January 2001, until 23 January 2022, for both the NYSE and the NASDAQ. We found that the *CSIE* market volatility estimator is consistently at least 10 times more sensitive to market changes, compared to the volatility estimate captured through the market indices. Furthermore, beta values confirm a consistently lower volatility risk for market indices overall, between 50% and 90% lower, compared to the volatility risk of the entire market in various time intervals and rolling windows.

## 1. Introduction

Uncertainty in stock markets is intimately related to time. The temporal dimension of uncertainty has traditionally been embedded in the volatility estimate of a given exchange-traded security over a certain time frame. Additionally, the historical volatility of the entire market can be derived from the volatility of a market index, which is more or less comprehensive and more or less representative for the market as a whole. The volatility dynamics of the securities market have challenged researchers and practitioners alike for decades; directions of interest range from the analysis of time series of individual market indices [1,2] to portfolio diversification (construction) strategies rooted in cross-sectional volatility models that aim to discriminate between market and idiosyncratic volatility and systemic and peculiar risks [3,4].

Cross-sectional volatility is defined as the dispersion of a set of stock returns over a time interval [5]. For example, if the standard deviation of the returns of stocks is small, it is concluded that the considered stocks behave similarly, and there is little opportunity to outperform the market [6,7]. In a contrasting scenario, high levels of cross-sectional volatility allow opportunities to be created to construct portfolios with significant differences in performance, as Ankrim and Ding (2002) argue in [8].

In their cross-sectional study [9], Fama and French (1992) emphasized the empirical contradiction that average returns on small-cap stocks are too high, given their beta estimate, and that average returns on large stocks are too low. Resorting to such beta estimates in cross-sectional studies is a natural and often-used approach, since it measures the volatility of an individual stock compared to the systematic risk of the entire market or, for simplification, the volatility risk of a market index. In statistical terms, the betas are the slopes of the line through a regression of data points for different periods. If the volatility of the market return is a systematic risk factor, the arbitrage pricing theory or a factor model predicts that aggregate volatility should also be priced in the cross section of stocks.

Studies of the stock market in such a cross section have mainly concerned portfolio construction, particularly on how to gauge the sensitivity of selected stocks to their expected returns, with respect to innovations in aggregate volatility [10,11].

For example, Ang et al. substantiated that such multifactor risk models predict that aggregate volatility should be a cross-sectional risk factor and consequently used changes in the VIX index of the Chicago Board Options Exchange (CBOE) to proxy innovations in aggregate volatility [12]. They investigated how the stochastic volatility of the market is priced in the cross section of expected stock returns and empirically tested the hypothesis that stocks with different sensitivities to innovations in aggregate volatility should have different expected returns. They maintain that volatility has to be priced in the cross section of stocks as well, should the volatility of market return be a systematic risk factor [13,14,15]. 

Many approaches to studying the market in the cross section revolve around exploring the information content of the cross-sectional dispersion of stock returns to predict volatility [16,17,18]. In particular, the research interest has been directed towards incorporating in volatility models the information provided by the dispersion of stock returns and testing whether this additional information provides, statistically speaking, more accurate forecasts. Forecasts could be related to the volatility of the entire market, industry, or sector-level volatility [19]. Another direction of studies concerns how cross-sectional dispersion in the returns of different stocks can help predict volatility of a market index, such as the S&P 500 [20].

Some cross-sectional investigations are also conducted to separate aggregate volatility risk from idiosyncratic volatility, in the context of portfolio selection, and particularly to hedge against individual or sector idiosyncratic risks, as in [21,22,23,24].

Studies in the cross section have also been conducted on the global stock market. Zunino et al. [25] analysed the price returns of 48 stock market indices from different countries with the aim of distinguishing the stage of each stock market development (developed, emerging, or frontier countries) by using the Tarnopolski model representation space [26]. The Tarnopolski representation space helps plot the number of turning points of a time series versus its associated Abbe values [27].

Our study did not necessarily aim to directly assist portfolio selection but rather to provide a comprehensive cross-sectional volatility estimator, constructed taking into account all the symbols listed and traded on a given market or a subset of symbols built based on the purpose for the study: sector, industry, localization, diversification, affinity, etc. To our best knowledge, there is no cross-sectional volatility estimator that

(a)Takes into account all the listed and traded symbols of a given market;(b)Includes in the model not only the daily OHLC prices, but also the traded volume.

This additional market information provided by the volume of transactions at a given price level has been recognized and highlighted by Ausloos and Ivanova [28,29]. They pioneered a generalized momentum indicator and subsequently proposed a moving average of the generalized momentum, which is volume-weighted. In doing so, they introduced some analogy to a generalized Brownian particle’s “time-dependent mass” [30]. The number of trades and the volume of transactions are measures that indicate investor considerations about the evolution of a given stock. In the 1970s, Philippatos and Wilson [31,32] considered entropy to be a better statistical measure of risk than variance, since entropy does not make assumptions about the underlying probability distribution. Horowitz and Horowitz [33] attempted to conclude that entropy measures do contain worthwhile information otherwise unavailable or rather unrecognized by standard statistical techniques, such as variance or correlation analysis. Recently, Zhou et al. in [34] reviewed concepts and the principles concerning entropy models for applications in the field of finance, especially in portfolio selection and asset valuation. Regarding the traded volume data, in [35] we introduced an intraday intrinsic entropy model as an indicator to dynamically gauge investors’ interest in a given exchange-traded security. Therefore, for a given stock, the ratio between daily volume traded and the overall volume traded in the period is intended to provide additional information on the history of the intercorrelations between the price and volume of transactions during the considered time interval.

In such a framework, the research questions of our study are:How does the cross-sectional intrinsic entropy (*CSIE*) market volatility estimator of the NYSE and NASDAQ stock markets relate to the volatility evolution of the corresponding indices, the S&P 500 Index (S&P500), Dow Jones Industrial Average (DJIA), and the NASDAQ Composite, respectively?Does the volatility of market indices follow the cross-sectional intrinsic entropy as a volatility estimator for the entire market?

To answer the laid-out research questions, we used an approximate 6000 days of reference points, starting with 1 January 2001, until 23 January 2022, for both the NYSE and the NASDAQ. The cross-sectional intrinsic entropy volatility estimator was computed on a daily basis for the entire data set of 5494 files, the equivalent of 21 years of daily trading data for the NYSE and NASDAQ. We tested the covariance between the historical volatility of the market indices S&P500, DJI, NASDAQ Composite, and the cross-sectional intrinsic entropy (*CSIE*) volatility estimator. Since the cross-sectional intrinsic entropy provides daily values, we conducted the tests for various time intervals by computing the corresponding moving averages for the cross-sectional intrinsic entropy volatility estimator. The comparison between historical volatility, based on time series, and the moving averages for the cross-sectional intrinsic entropy volatility estimator was made through multiple betas corresponding to multiple time intervals. These particular betas measure the risk rate associated with the volatility of the market indices relative to the volatility risk that accompanies the entire population of traded symbols provided by the cross-sectional intrinsic entropy market volatility estimator.

The remainder of the paper is organized as follows. In Section 2 we discuss the historical data used in the study (Section 2.1) and define the cross-sectional intrinsic entropy market volatility estimation model, along with its computational methodology (Section 2.2). In Section 3 we present the results of or study through applications on the NYSE and NASDAQ stock markets. Note that for readability purposes, we emphasize the NYSE in the main text and report figures, tables, and notes on the NASDAQ in the Appendices. Section 4 is dedicated to the analysis of the results, on the basis of which we provide answers to the research questions. In addition to the cross-sectional intrinsic entropy model that we propose, our presently main contribution to the field is provided from the results presented in this article that show that the *CSIE* market volatility estimator is consistently at least 10 times more sensitive to market changes, compared to the volatility estimate captured through the market indices for both the NYSE and the NASDAQ. This high variability of the market volatility estimate provided by the *CSIE* corroborates the lower volatility risk traditionally associated with the market indices. Additionally, we defined and calculated specific betas as ratios between the covariance of market index volatility and market volatility estimates provided by *CSIE* in relation to the variance of the cross-sectional intrinsic entropy of the market. The beta values confirm a consistently lower volatility risk for the market indices compared to the entire market on multiple time intervals and various rolling windows used for computing the volatility estimates: overall, between 50% and 90% lower volatility risk. Finally, in Section 5 we conclude our study and indicate future research directions on volatility forecasting using a cross-sectional intrinsic entropy model.

## 2. Materials and Methods

### 2.1. Discussion of the Historical Data Used in the Study

We conducted the study on end-of-day (EOD) data provided by eoddata.com—Historical Stock Data—End of Day Data (https://www.eoddata.com, accessed on 28 January 2022) for two securities markets: The New Your Stock Exchange (NYSE) and the National Association of Securities Dealers Automated Quotations (NASDAQ). The EOD data consist of a daily file containing the OHLC prices and volume for all the symbols listed and actually traded on the market on any given day for the last 20 years. In this respect, we have available for our research over 6000 days or reference points, starting on 1 January 2001, for both the NYSE and the NASDAQ. The actual file structure and the data it contains are presented in Table 1. The NYSE EOD data file for 21 January 2022, contains 3562 symbols; see Table 1 showing the first 12 and the last 12 of them. 

The EOD data for NASDAQ are organized in a similar way. It should be noted that the historical data files provided by eoddata.com are adjusted for splits, but do not have adjusted data for dividends. In the EOD data files, for both the NYSE and NASDAQ markets, prices are in USD. The daily trading volume results from the number of shares traded per day for each symbol.

It is relevant to note that the number of symbols listed and traded on a daily basis has evolved considerably over time. Figure 1 shows that about 1000 symbols were listed on the NYSE in the beginning of 2001, but in the beginning of 2022, over 3500 symbols were listed and traded. In Appendix A, Figure A1 shows the fact that on the NASDAQ, a higher rate of new listings has been recorded as well, from a few more than 750 symbols in the beginning of 2001 to more than 5100 listed and daily traded symbols in the beginning of 2022.

Considering the methodology of our investigation, which uses EOD data for all symbols listed on the market, this aspect must be taken into account. That is, the number of symbols traded daily can be assimilated to the number of microstates that the stock market, as an open system, may actually have. All of these microstates contribute to the daily cross-sectional intrinsic entropy of the market. For example, Figure 2 shows the distribution of the closing prices on the NYSE for 21 January 2022. 

We comment that, due to scaling reasons, we consider for this display only prices less than or equal to USD100 (3424 symbols out of 3562 that were traded on the day). We skip the closing prices above USD100 since they are recorded for less than 4% of the total number of traded symbols. 

Therefore, a legitimate question would be the following: Are the daily intrinsic entropy estimates comparable since the number of traded companies (microstates) may differ from day to day? In fact, historically, a continuous increase in the number of listed and traded symbols has generally been recorded on any given exchange. It is a natural process; more and more private companies reach the size, complexity of their operations, and financial needs that present the proposition of going public.

Securities markets are not isolated systems; symbols (companies) are listed, delisted, traded, and restricted from being traded as part of the process of having an open and dynamic marketplace. If we consider the listed companies as microstates within the stock exchange as an unclosed system, then the extensive property of the entropy still holds only if the constituents are distinguishable. This condition is in fact satisfied, since the system constituents are very precisely defined entities, and the analogy is consistent with the concept of adding new states (by coarse graining) to a system characterized by discernible microstates.

Within the market as a whole, listed companies (symbols) can hardly be clustered solely on the basis of the prices (OHLC) at which their stocks are traded on the exchange. They are traditionally grouped on the basis of their activity sector or other intimately peculiar fundamentals. Alternatively, we performed a cluster analysis using OHLC prices as variables; see Table 1, for example, with the intention of identifying potential similarities/differences between daily OHLC prices.

For hierarchical cluster analysis, we employed an agglomerative algorithm, using the average clustering method, along with the correlation between variables as metrics [36]. 

Figure 3 and Figure 4 show the clusters identified among the OHLC prices over four consecutive days: 1–3 and 6 December 2021.

It should be noted that on 1 and 3 December 2021, the NYSE S&P 500 index closed lower than the opening price, while on 2 and 6 December 2021, the index closed at a higher price than the opening price.

Therefore, the clusters identified among the OHLC prices reflect the similarities between Open and High, Close, and Low, respectively, for the days in which the index closed lower than its opening. On the contrary, for the days in which the S&P 500 closed higher than its opening, the clusters of OHLC prices reflect the similarities between Close and High, Open, and Low, respectively. A cluster analysis of OHLC prices revealed that similarities and differences among these prices can change on a daily basis, as a confirmation of the way in which trading activity is driven by investors’ actions throughout the day.

### 2.2. The Cross-Sectional Intrinsic Entropy (CSIE) Model and Methodology 

The *CSIE* model considers EOD data for various time intervals; therefore, the methodology must take into account that the data are distributed in multiple files, each file containing the EOD data for all the symbols that are traded in a given day $d_{i}$; $d_{i}\in$ $\left\{d_{1},\;\;d_{2},\;d_{3},\ldots ,\;d_{k},\ldots ,\;d_{n}\right\}$, where i is the number of days in the studied time interval, i∈ $\left\{1,\;2,\;3,\ldots ,\;n\right\}$. Table 2 offers a systematised perspective of the data used in the model studied.

For each symbol $x_{i}$ listed and traded in the day $d_{k}$, we have the following available:

-$x_{i}^{O}$—the open (O) price;-$x_{i}^{H}$—the high (H) price;-$x_{i}^{L}$—the low (L) price;-$x_{i}^{C}$—the close (C) price;-$x_{i}^{V}$—the traded volume (V).

The values $\left\{m_{1},\;m_{2},\;m_{3},\ldots ,\;m_{k},\ldots ,\;m_{n}\right\}$ represent the number of the symbols listed and traded on the market in the corresponding days $\left\{d_{1},\;\;d_{2},\;d_{3},\ldots ,\;d_{k},\ldots ,\;d_{n}\right\}$.

The total value traded daily, considered at the end of each trading day $d_{k}$, is given by the following relation:(1)$$S_{d_{k}}=\overset{m_{k}}{\underset{i=1}{\sum }}x_{i}^{C}x_{i}^{V},\;\;\mathrm{for}\;k\in \left[1,\;n\right],$$
$x_{i}^{C}$ and $x_{i}^{V}$ being the close price and the traded volume for the symbol $x_{i}$. Therefore, for each daily $x_{i}$, the daily weight of a symbol’s value in the overall traded value $S_{d_{k}}$ on day $d_{k}$ is defined as:(2)$$\psi_{d_{k}}^{x_{i}}=\frac{x_{i}^{C}x_{i}^{V}}{S_{d_{k}}},\;\;\mathrm{for}\;i\;\in \;\left[1,\;\;m_{k}\right],$$
where $\;m_{k}$ is the number of the symbols listed and traded on the market in the corresponding day $d_{k}$.

If we notate $s_{d_{k}}^{x_{i}}=x_{i}^{C}x_{i}^{V},\;\forall \;x_{i}\in \left\{x_{1},\;x_{2},\;x_{3},\ldots ,\;x_{\;m_{k}}\right\},\;for\;k\;\in \;\left[1,\;n\right]$, then
(3)$$\psi_{d_{k}}^{x_{i}}=\frac{s_{d_{k}}^{x_{i}}}{S_{d_{k}}},\;\;\mathrm{for}\;i\;\in \;\left[1,\;\;m_{k}\right].$$
Such ratios $\psi_{d_{k}}^{x_{i}}$ denote the portion of the traded value corresponding to the symbol $x_{i}$ in the overall value $S_{d_{k}}$, the total amount of money exchanged on the market on day $d_{k}$.

With the above notation, the cross-sectional intrinsic entropy model is the following.
(4)$$H_{d_{k}}=\left(1-f_{d_{k}}\right)H_{d_{k}}^{OC}+f_{d_{k}}H_{d_{k}}^{OLHC}$$
The components $H_{d_{k}}^{OC}$ and $H_{d_{k}}^{OLHC}$ are defined as follows:(5)$$H_{d_{k}}^{OC}=-\overset{m_{k}}{\underset{i=1}{\sum }}\left(\frac{x_{i}^{C}}{x_{i}^{O}}-1\right)\psi_{d_{k}}^{x_{i}}\ln\psi_{d_{k}}^{x_{i}}$$
(6)$$H_{d_{k}}^{OLHC}=-\overset{m_{k}}{\underset{i=1}{\sum }}\left[\left(\frac{x_{i}^{H}}{x_{i}^{O}}-1\right)\left(\frac{x_{i}^{H}}{x_{i}^{C}}-1\right)+\left(\frac{x_{i}^{L}}{x_{i}^{O}}-1\right)\left(\frac{x_{i}^{L}}{x_{i}^{C}}-1\right)\right]\psi_{d_{k}}^{x_{i}}\ln\psi_{d_{k}}^{x_{i}}$$
(7)$$f_{d_{k}}=\frac{\alpha -1}{\alpha +\frac{\;m_{k}+1}{\;m_{k}-1}}$$
$\forall \;d_{k}\in \left\{d_{1},\;\;d_{2},\;d_{3},\ldots ,\;d_{n}\right\}$, for $m_{k}\in \;\left\{m_{1},\;m_{2},\;m_{3},\ldots ,\;m_{n}\right\}$.

We comment that Equation (4) contains in the right-hand-side term a linear combination between the *CSIE* component weighted with the variation between the close the open prices $H_{d_{k}}^{OC}$ and the *CSIE* component weighted with OHLC variations $H_{d_{k}}^{OLHC}$, in the manner introduced for the intrinsic entropy (*IE*) volatility estimator [37].

The *IE* model is a time-series-based volatility estimator. It is conceived to use historical daily OHLC prices available for any exchange traded securities: stocks, market indices, commodities, ETFs, etc. The computation methodology is similar to all the others variance-based historical volatility estimators: classical close-to-close, Garman–Klass [38], Parkinson [39], Rogers–Satchell [40,41], Yang–Zhang [42].

The value of $f_{d_{k}}$ from (7) is consistent with the determination by Yang and Zhang. In their influential paper on drift-independent volatility estimation using OHLC prices, they searched for a value *k* (see Equation (11)) for which the variance of the volatility estimator reaches the minimum [42]. Based on the work of Rogers and Satchel [40], who showed that α ≤ 2 by using triangle inequality, Yang and Zhang calculated that α ≤ 1.5 for all drifts. When the drift is zero, α reaches the minimum value. Therefore, based on the Garman and Klass [38] formulas of moments, Yang and Zhang calculated the value 1.331 for α, when the drift is zero. To optimize their volatility estimator for situations exhibiting a small drift, Yang and Zhang suggested setting  α=1.34 in practice. Since the significance of the terms $H_{d_{k}}^{OC}$ and $H_{d_{k}}^{OLHC}$ is similar to that of $V_{OC}$ and $V_{RS}$ from the Yang–Zhang volatility estimator (11), we followed the same rationale for using α=1.34 to calculate the weight $f_{d_{k}}$.

The number of daily traded symbols, $m_{k}\in \left\{m_{1},\;m_{2},\;m_{3},\ldots ,\;m_{n}\right\}$, is not a stationary value (see Figure 1); hence, it has to be assessed for each day, $d_{k}\in \left\{d_{1},\;\;d_{2},\;d_{3},\ldots ,\;d_{n}\right\}$ from the considered time interval. Since we cannot rely on the close price from the previous day for each individual symbol, and since there is no guarantee that a given symbol is traded every single day in the considered time frame, the daily *CSIE* relies only on the symbols that are actually traded in that day and are contained in the corresponding daily file. Not only is the size of the file not a constant, but its set of symbols content can also vary.

Unlike volatility estimation using the *IE* model that we introduced in [37], the value $f_{d_{k}}$ was used for weighting the component $H_{d_{k}}^{OLHC}$, which quantifies the daily fine variations between the OHLC prices, while the difference $\left(1-f_{d_{k}}\right)$ was used for weighting $H_{d_{k}}^{OC}$, which accounts for the coarse variation between daily open and close prices.

## 3. Results

We now present our empirical results on the cross-sectional intrinsic entropy (*CSIE*) for the NYSE and the NASDAQ. The daily *CSIE* for the NYSE between 1 January 2001 and 21 January 2022 is shown in Figure 5.

Along with the daily evolution of *CSIE*, 5295 reference points, Figure 5 depicts the 60-day moving average of *CSIE* for the entire time interval.

Furthermore, Figure 6 emphasizes the evolution of the number of symbols traded on the NYSE, on a daily basis, for the entire time interval of available data used in our study, 21 years, between 1 January 2001 and 21 January 2022. This provides a perspective of the daily volatility estimates for the NYSE, together with the evolution of the number of listed and daily traded companies as microstates within the stock exchange.

We provide in Appendix A Figure A2, Figure A3 and Figure A4, which show the evolution of *CSIE* for the NASDAQ in the same time interval, from 1 January 2001, to 21 January 2022, and the same number of 5295 reference points.

We comment that the daily *CSIE* market volatility estimates exhibit significant day-to-day variability on both the NYSE and the NASDAQ. To capture trends of market volatility in shorter time windows, we computed moving averages of the *CSIE* for both the NYSE and the NASDAQ.

Figure 7 shows the 60-day moving average of the cross-sectional intrinsic entropy of the NYSE between 1 January 2001 and 21 January 2022. It is noticeable in Figure 7 that even smoothed through the moving average of 60 days, the variability of the *CSIE* is considerably higher compared to the corresponding historical volatility estimates of the SP&500 in the same time interval. In Figure 8, from top to bottom, we present the evolution of daily closing prices of the S&P500 stock market index, along with historical volatility estimates provided by intrinsic entropy (*IE*), Yang–Zhang, Rogers–Satchell, Garman–Klass, Parkinson, close-to-close (Raw) volatility estimators, and the traded volume.

We now present our empirical investigation into the *CSIE* stock market volatility estimator variability for the NYSE and the NASDAQ and relate it to the historical volatility estimates computed for the representative indices of the two markets: S&P500, DJIA, and the NASDAQ Composite index, respectively. For each estimator of historical volatility and for various time intervals, we computed the variance and mean as follows. The datasets for daily OHLC prices of the stock market indices for both the NYSE and the NASDAQ, along with the daily traded volume, were sourced from Yahoo! Finance (https://finance.yahoo.com/, accessed on 28 January 2022). These are time series of historical data, as opposed to cross-sectional data that we used in the model of the CSIE market volatility estimator.
(8)$$Var=\sigma_{\hat{V}}^{2}=\frac{1}{n}\;\overset{n}{\underset{i=1}{\sum }}{\left({\hat{V}}_{i}-\bar{\hat{V}}\right)}^{2}\;,\;\;\;Mean=\bar{\hat{V}}=\frac{1}{n}\;\overset{n}{\underset{i=1}{\sum }}{\hat{V}}_{i}\;.\;$$
To avoid cluttering the presentation, the historical volatility estimators that we consider for our analysis are the classical close-to-close estimator, along with the advanced volatility estimators Rogers–Satchell, Yang–Zhang, and the intrinsic entropy volatility estimator (*IE*). 

We present briefly the historical volatility estimators based on daily OHLC prices and traded volume using the following notation:

-$O_{i}$—the open price of day *i*;-$C_{i-1}$—the closing price of the previous day *i* − 1;-$H_{i}$—the high price of day *i*;-$L_{i}$—the low price of day *i*;-$C_{i}$—the closing price of day *i*;-$q_{i}$—the traded volume (number of index contracts) of day *i*.
for $i\;\in \;\left[1,\;n\right]$, *n* being the number of days in the time interval considered.

With this notation, we write the classical close-to-close volatility estimator,
(9)$$V_{CC}=\sqrt{\frac{1}{n}\overset{n}{\underset{i=1}{\sum }}ln\left(\frac{C_{i}}{C_{i-1}}\right),}$$
the Rogers–Satchell Volatility Estimator,
(10)$$V_{RS}=\sqrt{\frac{1}{n}\overset{n}{\underset{i=1}{\sum }}\left[ln\left(\frac{H_{i}}{O_{i}}\right)ln\left(\frac{H_{i}}{C_{i}}\right)+ln\left(\frac{L_{i}}{O_{i}}\right)ln\left(\frac{L_{i}}{C_{i}}\right)\right]},$$
and the Yang–Zhang volatility estimator.
(11)$$V_{YZ}=\sqrt{V_{CO}^{2}+k\;V_{OC}^{2}+\left(1-k\right)\;V_{RS}^{2}}.$$
where $V_{CO}^{2}$ is the overnight volatility, between the previous close and the current open, and $V_{OC}^{2}$ is the open-to-close volatility of the current trading day.
(12)$$V_{CO}^{2}=\frac{1}{n}\overset{n}{\underset{i=1}{\sum }}{\left[ln\left(\frac{O_{i}}{C_{i-1}}\right)-\mu_{CO}\right]}^{2}$$
(13)$$V_{OC}^{2}=\frac{1}{n}\overset{n}{\underset{i=1}{\sum }}{\left[ln\left(\frac{C_{i}}{O_{i}}\right)-\mu_{OC}\right]}^{2}$$
The values $\mu_{CO}=\frac{1}{n}\;\sum_{i=1}^{n}ln\left(\frac{O_{i}}{C_{i-1}}\right)$ and $\mu_{OC}=\frac{1}{n}\;\sum_{i=1}^{n}ln\left(\frac{C_{i}}{O_{i}}\right)$ are corresponding averages of the previous close to current open log-return and current open-to-close log-return in the considered time interval, respectively. Yang and Zhang chose the constant *k* to minimize the variance of the $V_{YZ}$ estimator:(14)$$k=\frac{0.34}{1.34+\frac{n+1}{n-1}}\;.$$
The intrinsic entropy (*IE*) volatility estimator:(15)$$H=\;H^{CO}+k\;\;H^{OC}+\left(1-k\right)\;\;H^{OHLC},$$
where the *IE* component $H^{CO}$ estimates the overnight volatility, the component $H^{OC}$ is the open-to-close *IE* estimates weighted with *k*, and the IE $H^{OHLC}$ represents the fine estimate of volatility during the day, weighted with (1 − *k*).

The *IE* components are written as follows:(16)$$H^{CO}=-\overset{n}{\underset{i=1}{\sum }}ln\left(\frac{O_{i}}{C_{i-1}}\right)\;p_{i-1}\;ln\;p_{i-1}\;,$$
(17)$$H^{OC}=-\overset{n}{\underset{i=1}{\sum }}ln\left(\frac{C_{i}}{O_{i}}\right)\;p_{i}\;ln\;p_{i}\;,$$
(18)$$H^{OHLC}=-\overset{n}{\underset{i=1}{\sum }}\left[ln\left(\frac{H_{i}}{O_{i}}\right)ln\left(\frac{H_{i}}{C_{i}}\right)+ln\left(\frac{L_{i}}{O_{i}}\right)ln\left(\frac{L_{i}}{C_{i}}\right)\right]\;p_{i}\;ln\;p_{i}\;,$$
(19)$$p_{1}=\frac{q_{1}}{Q}\;,p_{2}=\frac{q_{2}}{Q}\;,\;p_{3}=\frac{q_{3}}{Q}\;,\ldots ,\;p_{n}=\frac{q_{n}}{Q}\;\;\;,\;\overset{n}{\underset{i=1}{\sum }}p_{i}=1,\;\;\;Q=\overset{n}{\underset{i=1}{\sum }}q_{i}\;.$$
Table 3 presents the mean values of the volatility estimates of the S&P500, provided by the historical volatility estimators: close-to-close, Rogers–Satchell, Yang–Zhang, and intrinsic entropy (*IE*). We comment that the time perspective goes backwards, starting with 21 January 2022, all the way towards 1 January 2001, for various time intervals and rolling windows. In the case of the cross-sectional intrinsic entropy (*CSIE*) of the NYSE stock market, a window size equal to the indicated number of days was used to calculate the corresponding moving average.

We note that the mean values of the *CSIE* market volatility estimate exhibit an order of magnitude comparable to the historical volatility estimates of S&P500 provided by the variance-based estimators: close-to-close, Rogers–Satchell, and Yang–Zhang. Intrinsic entropy estimates of the volatility of the S&P500 volatility were consistently lower due to the weight of the embedded traded volume. In Appendix B, Table A1 presents the mean values of the volatility estimates of the NASDAQ Composite for similar time intervals and rolling windows. In the case of *CSIE* of the NASDAQ stock market, a window size equal to the indicated number of days was used to calculate the corresponding moving average.

Regarding the variability comparison, we present in Table 4 the variance of the volatility estimates of the S&P500 for various time intervals and rolling windows. For the *CSIE* of the NYSE stock market, a window size equal to the indicated number of days was used to calculate the corresponding moving average.

We point out that the variance exhibited by the volatility estimates provided by *CSIE* of the NYSE stock market was consistently higher, with at least one order of magnitude, than the estimates provided by the volatility estimators for SP&500, for all time intervals and rolling windows/moving averages. As a statistical measure of the dispersion of data points around the mean, the variance figures exhibited by the cross-sectional intrinsic entropy estimate of market volatility show that *CSIE* captures volatility jumps that are not reflected in the historical volatility estimates of the corresponding market indices.

In Appendix B, Table A2 presents the variance of the volatility estimates of the NASDAQ Composite for similar time intervals and rolling windows/moving averages. The result data confirm that the volatility estimates provided by *CSIE* of the NASDAQ stock market are consistently higher, with at least one order of magnitude, than the estimates provided by the volatility estimators for the NASDAQ Composite index, for all time intervals and rolling windows/moving averages. Comparative mean and variance results show that the *CSIE* of both the NYSE and the NASDAQ exhibits a significantly higher sensitivity to changes in volatility at the market level when we take into account all traded companies, compared to the historical volatility estimates of the corresponding market indices.

## 4. Discussion

In addition to variance and mean as indicators that individually characterize volatility estimators, we tested the overall correlation between the CSIE volatility estimates and the volatility estimates of the market indices, namely the S&P500 of the NYSE. Estimators concerning historical volatility, close-to-close, Parkinson, Garman–Klass, Rogers–Satchell, and Yang–Zhang, were defined and computed based on the OHLC prices time series of market indices. The intrinsic entropy (IE) volatility estimator also used the daily OHLC prices of the market indices, along with the daily traded volume. 

In contrast to the time series used for historical volatility estimates of market indices, the CSIE market volatility estimator used cross-sectional data, that is, daily OHLC prices for all traded stocks on the market, to compute daily CSIE estimates. We now present the correlation between the volatility estimates for the S&P500 and the *CSIE* of the NYSE stock market, computed for multiple time intervals and rolling windows. In the case of the *CSIE* market volatility estimate of the NYSE, a window size equal to the indicated number of days was used to calculate the corresponding moving average. 

We underline that the weighting components of the entropic model in Equations (4)–(6) and (15)–(18) can generate positive values under the sum and, therefore, negative values for the *CSIE* market volatility estimate (see the mean values in Table 3, Figure 5, Figure 6 and Figure 7 and the following). It is the same feature exhibited by the IE on time series. We provide an interpretation of this feature in [31,33], which holds for all intrinsic entropy models (intraday *IE*, *IE* as historical volatility estimator, and *CSIE* market volatility estimator), namely: a negative value indicates a preponderantly sell movement in the market for the considered exchange-traded security, while a positive value indicates an inclination to buy.

To work with comparable entities, the *IE* volatility estimate of the market index and the cross-sectional intrinsic entropy, as a volatility estimate of the whole market, are computed in such a way as to produce positive values. Therefore, the relation (4) to calculate the *CSIE* estimate of market volatility becomes:(20)$$H_{d_{k}}=\left(1-f_{d_{k}}\right)\left|H_{d_{k}}^{OC}\right|+f_{d_{k}}\left|H_{d_{k}}^{OLHC}\right|$$
Similarly, departing from relation (15), the intrinsic entropy (*IE*) estimate of the market index volatility estimate becomes as follows:(21)$$H=\;\left|H^{CO}\right|+k\;\left|H^{OC}\right|+\left(1-k\right)\;\left|H^{OHLC}\right|$$
Table 5 presents the Pearson correlation coefficient values between the volatility estimates for the S&P500 and the *CSIE* of the NYSE stock market. In the case of the *CSIE* of the NYSE stock market, a window size equal to the indicated number of days was used to calculate the corresponding moving average. The detailed results presented in Table 5 are shown in Figure 9.

Overall, we noticed a consistent correlation between the *CSIE* volatility estimate of the NYSE market and the historical volatility estimates of the S&P500, for all considered time intervals and rolling windows/moving averages.

The least correlation with the *CSIE* volatility estimates of the NYSE market is shown by the volatility estimate of the S&P500 provided by the intrinsic entropy (*IE*). We comment that the *IE* estimates of the volatility of the S&P500 volatility are consistently lower due to the weight of the embedded traded volume. Consequently, the volume weighting in the *IE* volatility estimate, concerning the trading of S&P500, does not follow the volume weighting in the *CSIE* market volatility estimates, since it concerns the traded volume of each listed stock on the NYSE.

We now present the betas, defined ad hoc and computed as ratios between the covariances of market index volatility and market volatility estimates provided by *CSIE*, in relation to the variance of the cross-sectional intrinsic entropy of the market.
(22)$$\beta_{E,t,w}^{index}=\frac{Cov\left(V_{E,t,w}^{index},\;V_{t,w}^{market\;CSIE}\right)}{Var\left(V_{t,w}^{market\;CSIE}\right)}$$
where $V_{E,t,w}^{index}$ is the index volatility estimates, computed based on rolling windows of *w*-day, within the *t*-day time interval, and $V_{t,w}^{market\;CSIE}$ is volatility estimates of the entire market based on the cross-sectional intrinsic entropy and computed for the same *t*-day time interval. Index volatility estimates *E* are provided by the estimators, close-to-close, Parkinson, Garman–Klass, Rogers–Satchell, Yang–Zhang, and intrinsic entropies are defined and calculated based on historical OHLC daily prices of the S&P500 market index. We computed the beta values for time intervals of 30, 60, 120, 260, 520, 780, 1300, and 5295 days and rolling windows/moving averages of 5, 10, 20, and 30 days.

Table 6 presents the beta of the volatility estimates for the S&P500, relative to the *CSIE* of the NYSE stock market. In the case of the cross-sectional intrinsic entropy of the NYSE stock market, a window size equal to the indicated number of days was used to calculate the corresponding moving average. Figure 10 shows the beta values for the volatility estimates of the S&P500, relative to the *CSIE* of the NYSE stock market. We point out that for each time interval, there are four rolling windows/moving averages of 5, 10, 20, and 30 days (see Table 6).

We comment that the betas show, for all time intervals and rolling windows for the historical volatility estimates of the S&P500, that the associated volatility risk to the market index is generally significantly lower than the volatility risk associated to the entire NYSE market. Moreover, the volatility risk shows lower levels on shorter time intervals: over 90% lower for a 30-day period and over 75% lower for a 60-day period. We point out that the time perspective goes backward, starting with 21 January 2022, all the way towards 1 January 2001, for various time intervals and rolling windows.

## 5. Conclusions

The temporal dimension of the uncertainty of the stock market has traditionally been derived from the historical volatility of one or several market indices. This straightforward approach relies more or less on how comprehensive the index is constructed, and hence on how representative its volatility estimate is for the market as a whole. This paper introduces a cross-sectional estimation of stock market volatility based on the intrinsic entropy model. The proposed *CSIE* is defined and calculated as a daily value for the entire market, based on the daily traded OHLC prices, along with the daily traded volume for all symbols listed on The New York Stock Exchange (NYSE) and The National Association of Securities Dealers Automated Quotations (NASDAQ). 

We performe a comparative analysis between the time series obtained from the *CSIE* market volatility estimator and the historical volatility provided by the estimators close-to-close, Parkinson, Garman–Klass, Rogers–Satchell, Yang–Zhang, and intrinsic entropy (*IE*), defined and computed from historical OHLC daily prices of the Standard & Poor’s 500 index (S&P500), Dow Jones Industrial Average (DJIA), and the NASDAQ Composite index, respectively, for various time intervals. The empirical results presented in this article show that the *CSIE* market volatility estimator is consistently at least 10 times more sensitive to market changes than the volatility estimates captured through the market indices for both the NYSE and the NASDAQ. This high variability of the market volatility estimate provided by the *CSIE* corroborates the lower volatility risk traditionally associated with market indices. The results answer the first research question of our study—how does the cross-sectional intrinsic entropy (CSIE) market volatility estimator of the NYSE and NASDAQ stock markets relate to the volatility evolution of the corresponding indices, the S&P 500 Index (S&P500), Dow Jones Industrial Average (DJIA), and the NASDAQ Composite, respectively?

In order to answer the second research question of our study—does the volatility of market indices follow the cross-sectional intrinsic entropy as a volatility estimator for the entire market?—we define and calculate specific betas. These ad hoc defined betas are computed as ratios between the covariance of market index volatility and market volatility estimates provided by CSIE and the variance of the cross-sectional intrinsic entropy of the market. The beta values confirm a consistently lower volatility risk for the market indices compared to the entire market on the various lengths so investigated time intervals and various rolling windows used for computing the volatility estimates: overall between 50% and 90% lower volatility risk.

Furthermore, the *CSIE* volatility estimate provides a novel and comprehensive perspective on the evolution of volatility for the entire stock market. For example, Figure 11 shows the evolution of the cross-sectional intrinsic entropy of the NYSE between 1 June 2018 and 21 January 2022, along with its moving average of 30 days. 

To have a comparison with the results shown in Figure 6, the diameters of the bubbles highlight the evolution of the number of symbols traded daily.

Additionally, we plotted only the 30-day moving average of CSIE for the NYSE (Figure 12) and compared it with the evolution of the S&P500 index and its IE volatility estimate for the same time interval, between 1 June 2018 and 21 January 2022 (Figure 13).

We point out that, while the S&P500 prices had roughly a continuous ascension after the low reached at the beginning of the SARCOV-19 pandemic in early March 2020, the cross-sectional intrinsic entropy market volatility estimate emphasizes a significantly different evolution of the whole NYSE market’s volatility: there was a peak around the end of January 2021, followed by a continuous downward trend since then, until 21 January 2022, the last day we included in our study.

The *CSIE* market volatility estimate model may have multiple applications that have not been explored in the current study:(a)It can be employed for volatility estimate of portfolios of various number of exchange-traded securities, providing a convenient and effective manner to make comparisons to the volatility of the entire market or market indices;(b)The *CSIE* market volatility estimate can rely on various approaches when it comes to grouping companies listed on the market, at the sector level or in industries.

We comment that for short to medium time intervals and narrow windows for moving averages, the *CSIE* market volatility estimate can capture insightful perspectives of the market, which may provide a novel ground for volatility forecasting through GARCH-like models. For example, the evolution of the cross-sectional intrinsic entropy of the NYSE between 17 September 2021 and 21 January 2022 shows intense market volatility accompanied by not-negligible variability in the daily traded value of the market (Figure 14). Unlike Figure 6 and Figure 11, in Figure 14 the diameter of the bubbles denotes the daily market value.

In addition to the market volatility estimate in absolute terms, the peculiar property of the intrinsic entropy model of allowing negative values for the volatility estimate is present in the cross-sectional intrinsic entropy of the stock market as well. Similarly, in Figure 7 and Figure 12, Figure 15 depicts the five-day moving average of the cross-sectional intrinsic entropy of the NYSE between 17 September 2021 and 21 January 2022, and emphasizes market volatility associated with preponderant selling activity.

## Figures and Tables

**Figure 1 entropy-24-00623-f001:**
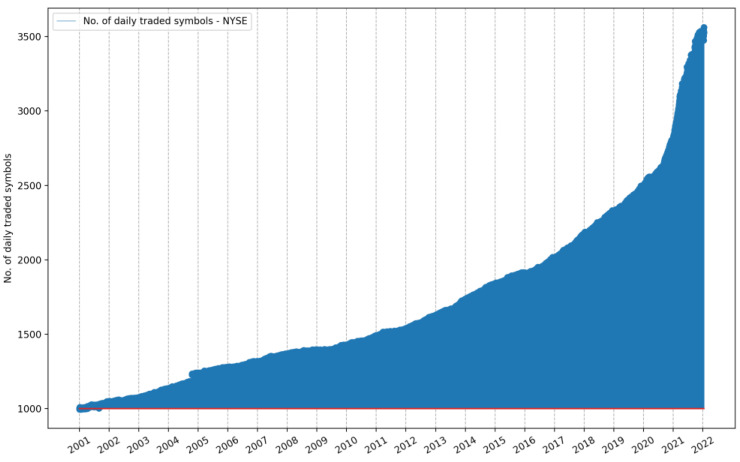
Evolution of the number of symbols traded on the NYSE over the last 21 years: from 1 January 2001, to 21 January 2022. The red horizontal line marks the number of 1008 symbols listed on 1 January 2001.

**Figure 2 entropy-24-00623-f002:**
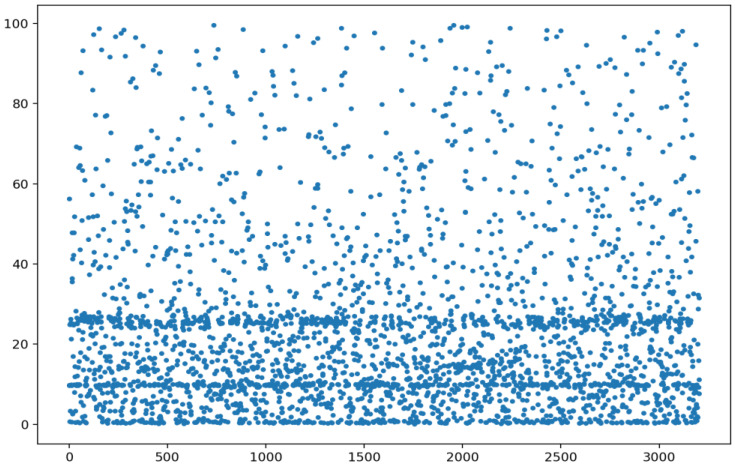
The distribution of closing prices on the NYSE: 21 January 2022—only prices less than or equal to USD100 (3424 symbols out of 3562) are displayed. Closing prices above USD100 are skipped from this graphical representation for readability purposes.

**Figure 3 entropy-24-00623-f003:**
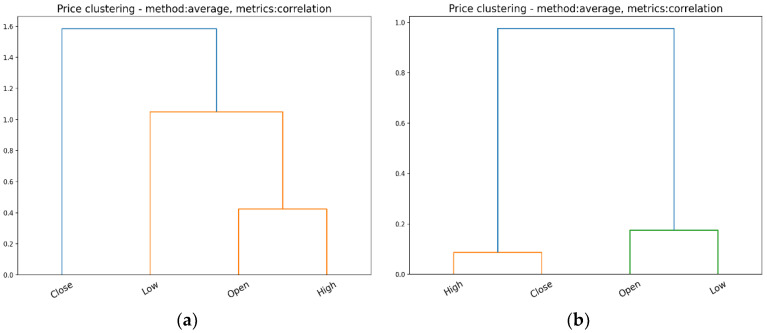
The hierarchical clustering among daily prices on NYSE: (**a**) 1 December 2021; (**b**) 2 December 2021.

**Figure 4 entropy-24-00623-f004:**
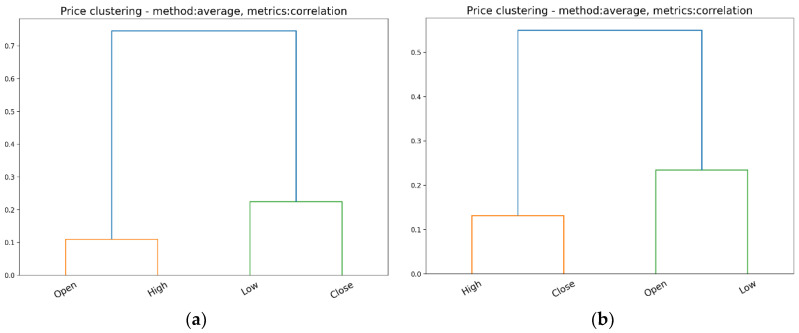
The hierarchical clustering among daily prices on NYSE: (**a**) 3 December 2021; (**b**) 6 December 2021.

**Figure 5 entropy-24-00623-f005:**
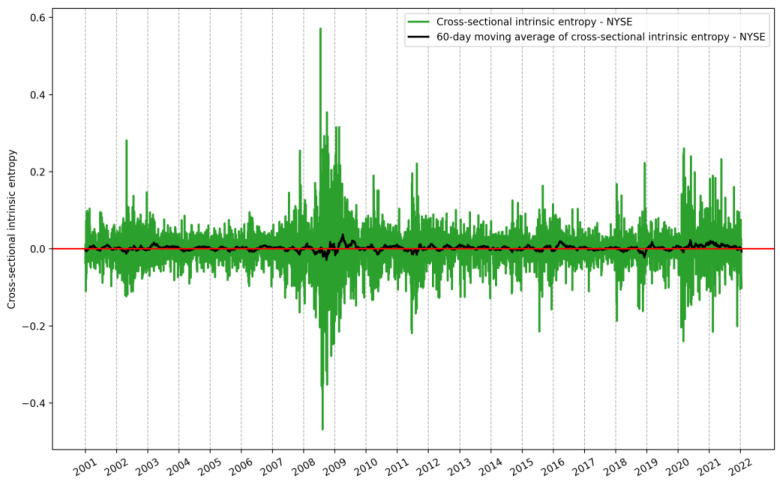
Evolution of the cross-sectional intrinsic entropy for the NYSE between 1 January 2001 and 21 January 2022, along with its moving average of 60 days.

**Figure 6 entropy-24-00623-f006:**
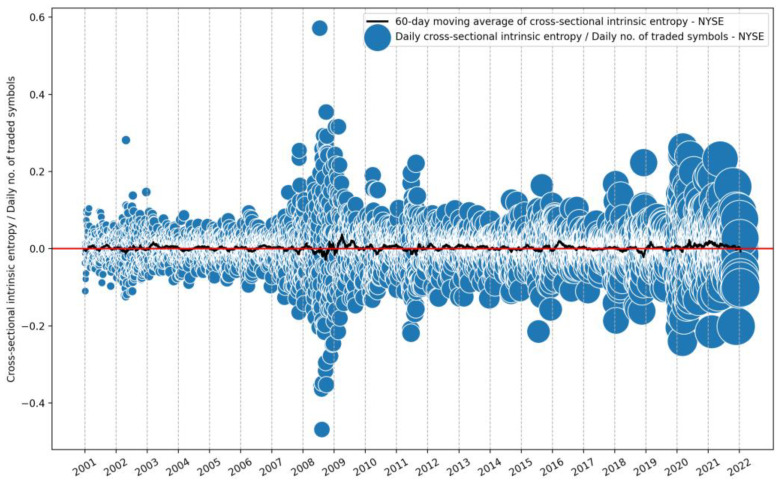
Evolution of cross-sectional intrinsic entropy for the NYSE between 1 January 2001 and 21 January 2022, along with its 60-day moving average. The diameter of the bubbles highlights the evolution of the number of symbols traded daily.

**Figure 7 entropy-24-00623-f007:**
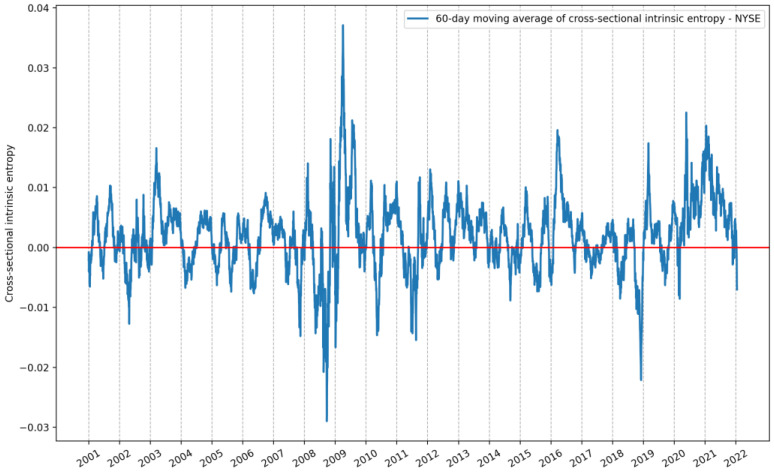
The 60-day moving average of the cross-sectional intrinsic entropy of the NYSE between 1 January 2001 and 21 January 2022.

**Figure 8 entropy-24-00623-f008:**
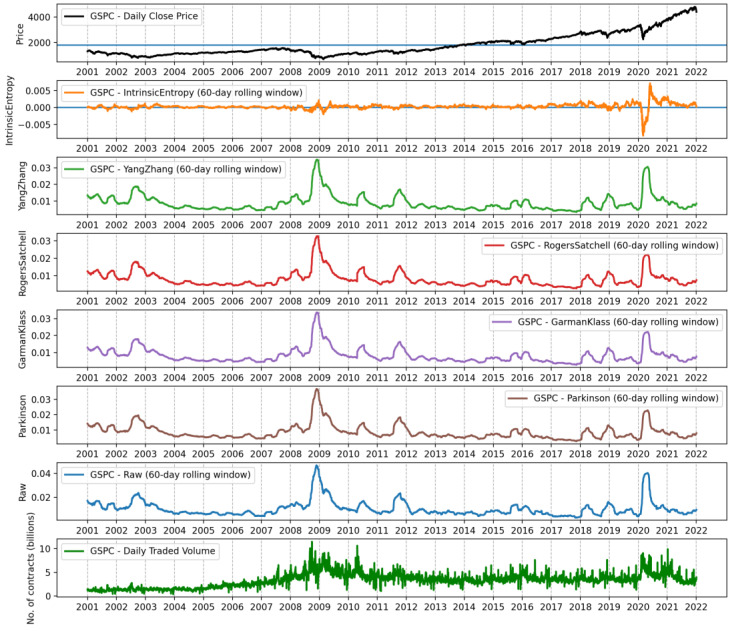
Sixty-day rolling window, from the top: intrinsic entropy-based estimates, Yang–Zhang, Rogers–Satchell, Garman–Klass, Parkinson, and close-to-close (Raw) volatility estimates for the S&P500 stock market index between 1 January 2001 and 21 January 2022.

**Figure 9 entropy-24-00623-f009:**
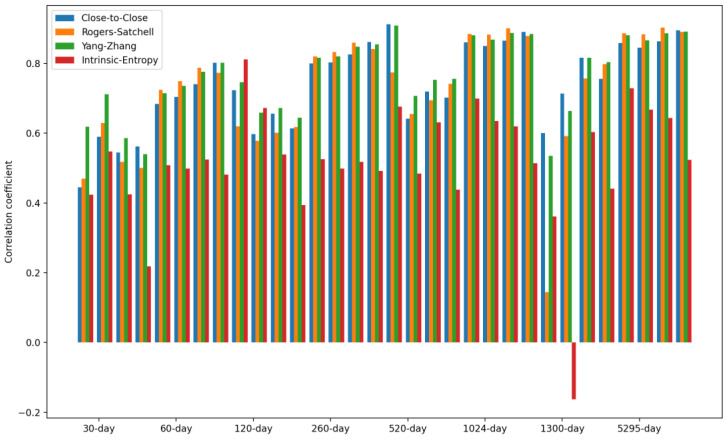
Pearson’s correlation coefficient between volatility estimates for the S&P500, and the cross-sectional intrinsic entropy of the NYSE stock market. For each time interval, there are four windows of 5, 10, 20, and 30 days (see Table 6).

**Figure 10 entropy-24-00623-f010:**
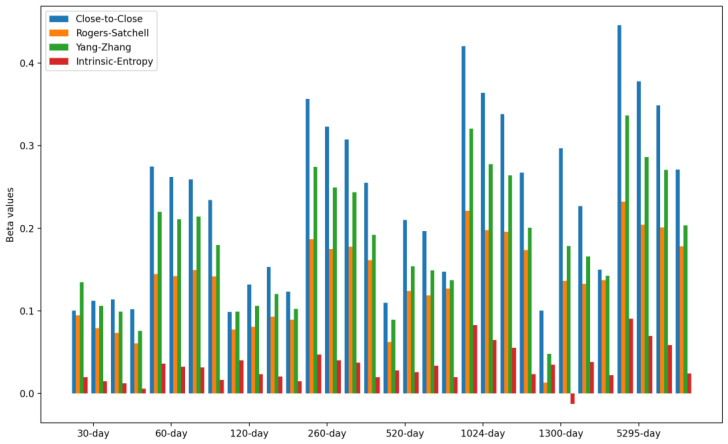
Beta of volatility estimates for the S&P500, relative to the cross-sectional intrinsic entropy of the NYSE stock market. For each time interval, there are four windows of 5, 10, 20, and 30 days (see Table 6).

**Figure 11 entropy-24-00623-f011:**
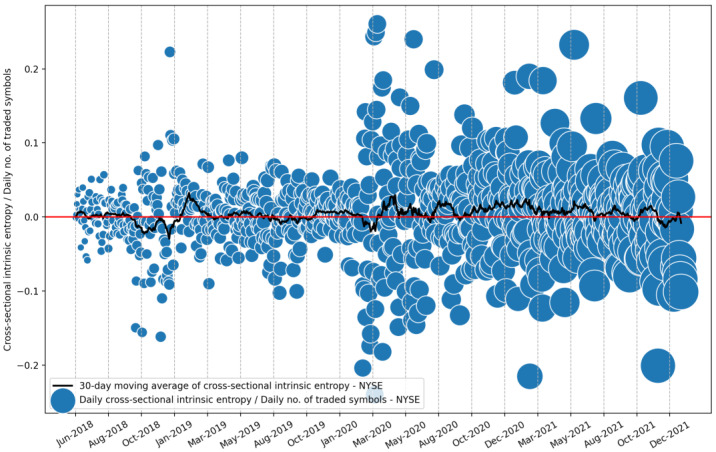
Evolution of cross-sectional intrinsic entropy of the NYSE between 1 June 2018 and 21 January 2022, along with its 30-day moving average. The diameters of the bubbles highlight the evolution of the number of symbols traded daily.

**Figure 12 entropy-24-00623-f012:**
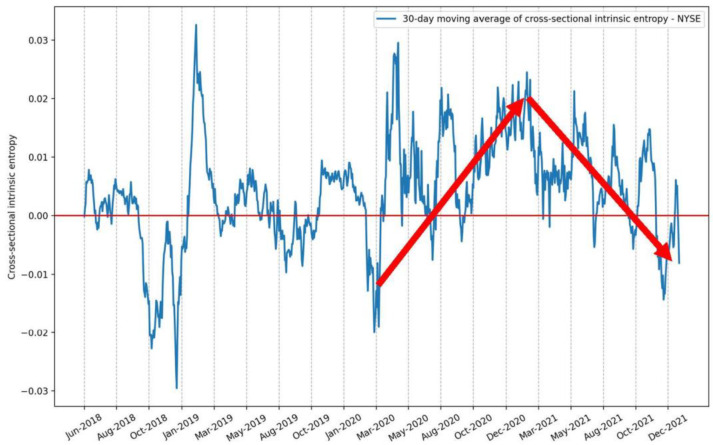
The 30-day moving average of the cross-sectional intrinsic entropy of the NYSE between 1 June 2018 and 21 January 2022. The *CSIE* volatility estimator for the entire NYSE market shows that there was a peak around the end of January 2021, followed by a continuous downward trend since then, until 21 January 2022.

**Figure 13 entropy-24-00623-f013:**
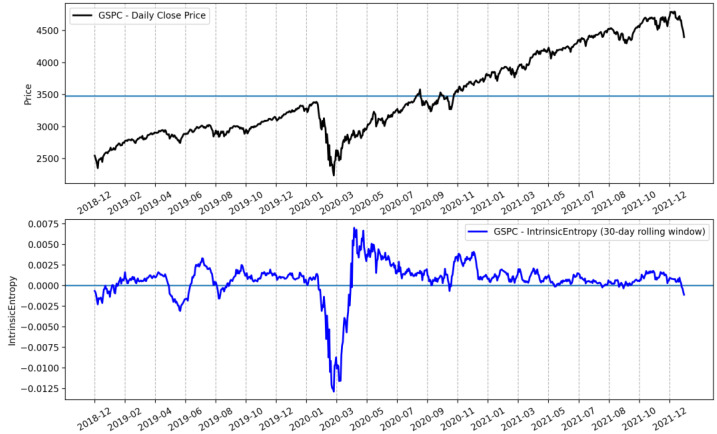
The 30-day rolling window of intrinsic entropy volatility estimates for the S&P500 stock market index between 1 June 2018 and 21 January 2022.

**Figure 14 entropy-24-00623-f014:**
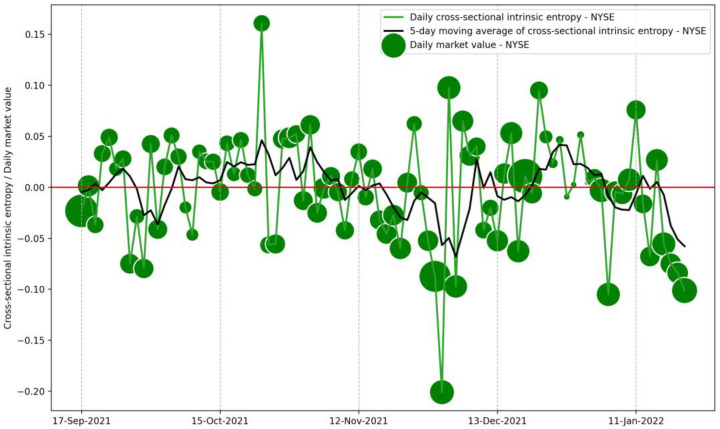
Evolution of the cross-sectional intrinsic entropy of the NYSE between 17 September 2021 and 21 January 2022, along with its 5-day moving average. The diameter of the bubbles denotes the daily value of the market.

**Figure 15 entropy-24-00623-f015:**
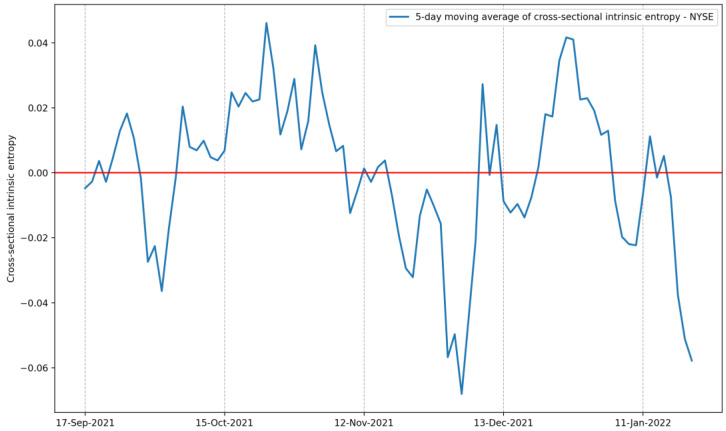
The 5-day moving average of the cross-sectional intrinsic entropy of the NYSE between 17 September 2021 and 21 January 2022.

**Table 1 entropy-24-00623-t001:** The beginning and end of the file that contains end-of-day (EOD) trading data for NYSE on 21 January 2022. The first and last 12 symbols are listed to provide an example of the data sample studied. Prices are in USD, and volume is the number of shares traded throughout the day.

No.	Symbol	Open	High	Low	Close	Volume
1	A	139.54	140.49	137.49	137.51	1,878,600
2	AA	60.02	60.15	56.04	56.21	11,024,900
3	AAC	9.74	9.74	9.72	9.74	1,164,800
4	AAC.U	9.86	9.89	9.84	9.84	45,900
5	AAC.W	0.7202	0.7629	0.6122	0.668	336,900
6	AAI-B	24.88	24.9	24.85	24.9	600
7	AAI-C	25.1	25.1011	24.83	25.04	3100
8	AAIC	3.49	3.49	3.4	3.41	142,400
9	AAIN	25.14	25.14	24.92	24.92	2100
10	AAM-A	25.39	25.49	25.32	25.32	156,700
11	AAM-B	26.75	26.75	26.239	26.26	169,400
12	AAN	21.33	22.24	21	21.32	259,300
…	…	…	…	…	…	…
3551	ZH	4.38	4.48	4.15	4.21	1,606,700
3552	ZIM	60.5	61.17	57.1	58.12	5,136,500
3553	ZIP	21.11	21.22	19.87	20.04	739,200
3554	ZME	1.78	1.8999	1.69	1.78	156,900
3555	ZNH	33	33.29	32.28	32.28	12,700
3556	ZTO	29.19	29.24	28.33	28.76	3,125,200
3557	ZTR	9.46	9.48	9.25	9.32	191,700
3558	ZTS	203.11	203.71	200.28	200.33	2,632,900
3559	ZUO	16.18	16.79	15.96	15.96	1,817,700
3560	ZVIA	7.5	8.02	7.4	7.64	168,900
3561	ZWS	32.39	32.39	31.4	31.51	874,600
3562	ZYME	11.43	11.7	11	11.21	953,700

**Table 2 entropy-24-00623-t002:** Daily data (open, high, low, and close prices along with the traded volume) for each listed symbol $x_{i}$ and trading day $t_{k}$.

Day	Symbol	Open	High	Low	Close	Volume
$d_{1}$	$x_{1}$	…				
	$x_{2}$		…			
	$x_{3}$			…		
	…	…	…	…	…	…
	$x_{m_{1}}$					…
$d_{2}$	$x_{1}$	…				
	$x_{2}$		…			
				…		
	…	…	…	…	…	…
	$x_{m_{2}}$					…
…						
$d_{k}$	$x_{1}$	$x_{1}^{O}$	$x_{1}^{H}$	$x_{1}^{L}$	$x_{1}^{C}$	$x_{1}^{V}$
	$x_{2}$	$x_{2}^{O}$	$x_{2}^{H}$	$x_{2}^{L}$	$x_{2}^{C}$	$x_{2}^{V}$
	…	…	…	…	…	…
	$x_{m_{k}}$	$x_{m_{k}}^{O}$	$x_{m_{k}}^{H}$	$x_{m_{k}}^{L}$	$x_{m_{k}}^{C}$	$x_{m_{k}}^{V}$
…						
$d_{n}$	$x_{1}$	…				
	$x_{2}$		…			
	$x_{3}$			…		
	…	…	…	…	…	…
	$x_{m_{n}}$					…

**Table 3 entropy-24-00623-t003:** Mean values of the volatility estimate of the S&P500, for various time intervals and rolling windows. For the cross-sectional intrinsic entropy of the NYSE stock market, a window size equal to the indicated number of days was used to calculate the corresponding moving average.

Time Interval (Days)	Rolling Window/Moving Averages (Days)	Close-to-Close (S&P500)	Rogers–Satchell (S&P500)	Yang–Zhang (S&P500)	Intrinsic Entropy (S&P500)	Cross-Sectional Intrinsic Entropy (NYSE Stock Market)
30	5	0.00943	0.00736	0.00916	−0.00010	−0.00042
60	5	0.00825	0.00670	0.00842	0.00023	−0.00390
120	5	0.00734	0.00607	0.00755	0.00025	0.00115
260	5	0.00748	0.00615	0.00762	0.00031	0.00507
520	5	0.01156	0.00835	0.01143	0.00042	0.00644
780	5	0.01016	0.00754	0.01026	0.00038	0.00520
1300	5	0.00850	0.00654	0.00879	0.00035	0.00268
5295	5	0.00969	0.00755	0.00914	0.00013	0.00197
30	10	0.01031	0.00779	0.00910	0.00011	0.00031
60	10	0.00846	0.00663	0.00785	0.00042	−0.00148
120	10	0.00758	0.00615	0.00718	0.00038	0.00185
260	10	0.00783	0.00631	0.00734	0.00047	0.00550
520	10	0.01194	0.00851	0.01098	0.00057	0.00662
780	10	0.01054	0.00772	0.00988	0.00053	0.00538
1300	10	0.00883	0.00668	0.00844	0.00049	0.00273
5295	10	0.00999	0.00768	0.00876	0.00019	0.00198
30	20	0.01084	0.00782	0.00905	0.00039	−0.00149
60	20	0.00853	0.00657	0.00762	0.00079	0.00101
120	20	0.00777	0.00614	0.00701	0.00053	0.00265
260	20	0.00802	0.00642	0.00726	0.00067	0.00611
520	20	0.01228	0.00867	0.01090	0.00070	0.00681
780	20	0.01093	0.00792	0.00987	0.00064	0.00564
1300	20	0.00912	0.00683	0.00838	0.00061	0.00277
5295	20	0.01018	0.00779	0.00865	0.00023	0.00199
30	30	0.01055	0.00769	0.00886	0.00065	−0.00508
60	30	0.00876	0.00670	0.00770	0.00096	0.00061
120	30	0.00779	0.00611	0.00694	0.00062	0.00232
260	30	0.00802	0.00648	0.00726	0.00076	0.00618
520	30	0.01253	0.00879	0.01098	0.00077	0.00680
780	30	0.01121	0.00808	0.00998	0.00068	0.00571
1300	30	0.00929	0.00693	0.00843	0.00067	0.00273
5295	30	0.01031	0.00787	0.00866	0.00026	0.00198

**Table 4 entropy-24-00623-t004:** Variance of the volatility estimates of the S&P500, for various time intervals and rolling windows. For the cross-sectional intrinsic entropy of the NYSE stock market, a window size equal to the indicated number of days was used to calculate the corresponding moving average.

Time Interval (Days)	Rolling Window/Moving Averages (days)	Close-to-Close (S&P500)	Rogers–Satchell (S&P500)	Yang–Zhang (S&P500)	Intrinsic Entropy (S&P500)	Cross-Sectional Intrinsic Entropy (NYSE Stock Market)
30	5	0.00000759	0.00000592	0.00000696	0.00000114	0.00057896
60	5	0.00001730	0.00000758	0.00001067	0.00000101	0.00062291
120	5	0.00001468	0.00000680	0.00000966	0.00000079	0.00047097
260	5	0.00001604	0.00000718	0.00000962	0.00000089	0.00047996
520	5	0.00015079	0.00003773	0.00008895	0.00000619	0.00064512
780	5	0.00011313	0.00003009	0.00006726	0.00000470	0.00049370
1300	5	0.00008053	0.00002380	0.00005021	0.00000320	0.00044034
5295	5	0.00006418	0.00002518	0.00003782	0.00000109	0.00049417
30	10	0.00000404	0.00000281	0.00000332	0.00000124	0.00027476
60	10	0.00001236	0.00000506	0.00000667	0.00000083	0.00033828
120	10	0.00001000	0.00000440	0.00000588	0.00000069	0.00023856
260	10	0.00000990	0.00000523	0.00000627	0.00000080	0.00019848
520	10	0.00013097	0.00003460	0.00007445	0.00000654	0.00027691
780	10	0.00009804	0.00002763	0.00005633	0.00000507	0.00021626
1300	10	0.00007029	0.00002181	0.00004200	0.00000336	0.00021329
5295	10	0.00005541	0.00002323	0.00003182	0.00000113	0.00024167
30	20	0.00000148	0.00000066	0.00000099	0.00000060	0.00012659
60	20	0.00000831	0.00000276	0.00000367	0.00000055	0.00018233
120	20	0.00000635	0.00000251	0.00000335	0.00000044	0.00011421
260	20	0.00000526	0.00000354	0.00000395	0.00000039	0.00009808
520	20	0.00011745	0.00003120	0.00006522	0.00000769	0.00012978
780	20	0.00008751	0.00002478	0.00004916	0.00000584	0.00010777
1300	20	0.00006315	0.00001968	0.00003686	0.00000370	0.00010785
5295	20	0.00004976	0.00002152	0.00002834	0.00000119	0.00012040
30	30	0.00000053	0.00000016	0.00000016	0.00000038	0.00002969
60	30	0.00000448	0.00000143	0.00000191	0.00000035	0.00007546
120	30	0.00000407	0.00000165	0.00000219	0.00000032	0.00005117
260	30	0.00000362	0.00000267	0.00000284	0.00000031	0.00005592
520	30	0.00011057	0.00002859	0.00005993	0.00000720	0.00007168
780	30	0.00008181	0.00002254	0.00004499	0.00000533	0.00006723
1300	30	0.00005956	0.00001816	0.00003410	0.00000334	0.00006915
5295	30	0.00004683	0.00002042	0.00002657	0.00000106	0.00007920

**Table 5 entropy-24-00623-t005:** Pearson’s correlation coefficient between volatility estimates for the S&P500 and the cross-sectional intrinsic entropy of the NYSE stock market. For the cross-sectional intrinsic entropy of the NYSE stock market, a window size equal to the indicated number of days was used to calculate the corresponding moving average.

Time Interval (Days)	Rolling Window/Moving Averages (days)	Close-to-Close (S&P500) Relative to Cross-Sectional Intrinsic Entropy	Rogers–Satchell (S&P500) Relative to Cross-Sectional Intrinsic Entropy	Yang–Zhang (S&P500) Relative to Cross-Sectional Intrinsic Entropy	Intrinsic Entropy (S&P500) Relative to Cross-Sectional Intrinsic Entropy
30	5	0.445	0.470	0.619	0.423
60	5	0.589	0.629	0.712	0.547
120	5	0.545	0.518	0.586	0.425
260	5	0.562	0.500	0.540	0.218
520	5	0.684	0.724	0.715	0.508
780	5	0.704	0.749	0.735	0.498
1300	5	0.740	0.787	0.776	0.525
5295	5	0.802	0.774	0.802	0.481
30	10	0.723	0.619	0.746	0.811
60	10	0.597	0.578	0.659	0.672
120	10	0.656	0.602	0.673	0.539
260	10	0.613	0.618	0.645	0.394
520	10	0.800	0.820	0.816	0.526
780	10	0.803	0.833	0.820	0.498
1300	10	0.826	0.860	0.848	0.517
5295	10	0.861	0.841	0.855	0.492
30	20	0.912	0.774	0.908	0.676
60	20	0.642	0.655	0.707	0.484
120	20	0.719	0.694	0.753	0.631
260	20	0.702	0.742	0.756	0.438
520	20	0.860	0.884	0.881	0.699
780	20	0.849	0.883	0.868	0.634
1300	20	0.865	0.900	0.887	0.619
5295	20	0.890	0.879	0.885	0.514
30	30	0.601	0.144	0.536	0.362
60	30	0.714	0.592	0.664	−0.163
120	30	0.817	0.757	0.817	0.603
260	30	0.756	0.798	0.804	0.441
520	30	0.858	0.886	0.880	0.729
780	30	0.845	0.883	0.866	0.667
1300	30	0.863	0.903	0.886	0.643
5295	30	0.895	0.890	0.891	0.524

**Table 6 entropy-24-00623-t006:** Beta of volatility estimates for the S&P500, relative to the cross-sectional intrinsic entropy of the NYSE stock market. For the cross-sectional intrinsic entropy of the NYSE stock market, a window size equal to the indicated number of days was used to calculate the corresponding moving average.

Time Interval (Days)	Rolling Window/Moving Averages (Days)	Close-to-Close (S&P500) Relative to Cross-Sectional Intrinsic Entropy	Rogers–Satchell (S&P500) Relative to Cross-Sectional Intrinsic Entropy	Yang–Zhang (S&P500) Relative to Cross-Sectional Intrinsic Entropy	Intrinsic Entropy (S&P500) Relative to Cross-Sectional Intrinsic Entropy
30	5	0.1002	0.0947	0.1349	0.0197
60	5	0.1121	0.0793	0.1060	0.0150
120	5	0.1138	0.0736	0.0993	0.0125
260	5	0.1021	0.0608	0.0759	0.0059
520	5	0.2748	0.1447	0.2201	0.0360
780	5	0.2623	0.1422	0.2111	0.0326
1300	5	0.2592	0.1496	0.2142	0.0315
5295	5	0.2343	0.1416	0.1799	0.0167
30	10	0.0989	0.0777	0.0992	0.0404
60	10	0.1319	0.0809	0.1060	0.0234
120	10	0.1531	0.0932	0.1204	0.0207
260	10	0.1235	0.0894	0.1025	0.0151
520	10	0.3567	0.1868	0.2743	0.0474
780	10	0.3231	0.1749	0.2493	0.0404
1300	10	0.3075	0.1779	0.2436	0.0375
5295	10	0.2551	0.1615	0.1922	0.0197
30	20	0.1099	0.0624	0.0893	0.0282
60	20	0.2100	0.1240	0.1541	0.0260
120	20	0.1965	0.1189	0.1492	0.0337
260	20	0.1474	0.1271	0.1372	0.0199
520	20	0.4206	0.2212	0.3205	0.0830
780	20	0.3642	0.1978	0.2778	0.0649
1300	20	0.3382	0.1960	0.2642	0.0553
5295	20	0.2676	0.1738	0.2009	0.0237
30	30	0.1006	0.0131	0.0479	0.0351
60	30	0.2969	0.1363	0.1785	−0.0125
120	30	0.2271	0.1329	0.1661	0.0384
260	30	0.1501	0.1371	0.1427	0.0224
520	30	0.4459	0.2322	0.3364	0.0904
780	30	0.3779	0.2043	0.2862	0.0697
1300	30	0.3488	0.2011	0.2706	0.0586
5295	30	0.2709	0.1781	0.2035	0.0242

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
