# Peer review of "The Cross-Sectional Intrinsic Entropy—A Comprehensive Stock Market Volatility Estimator"

_entropy, 2022, doi:10.3390/e24050623_

Round 1
Reviewer 1 Report
The authors presents an empirical study of the American stock market by means of the cross-sectional intrinsic entropy. Information theory related measures are gaining momentum in econophysics. In this sense I recommend also to cite alternative approaches as those in the following paper:
Zunino, L., Olivares, F., Bariviera, A. F., & Rosso, O. A. (2017). A simple and fast representation space for classifying complex time series. Physics Letters A, 381(11), 1021–1028. https://doi.org/10.1016/j.physleta.2017.01.047Author Response
Thank you for your suggestions!

Reviewer 2 Report
In the manuscript “The cross-sectional intrinsic entropy…”, its authors introduce a new measure of stock market volatility based on entropy. It has been inspired by a well-known concept of intrinsic entropy, but here used in a cross-sectional manner. The manuscript is interesting, complies with the Journal’s scope, and the new measure has large potential to be broadly used in econometrics. It could have been more concisely written (there are parts, which are too detailed or repetitive, and, thus, unnecessary - e.g., Tables 1 & 2), but the overall impression is favourable. I do not have any comments that would require a major revision, but several minor flaws are listed below.
- Tick mark labels in all figures have to be substantially increased, because they are difficult to read even in an A4 hardcopy.
- The numbers in all tables are given with excessive precision. Please reduce it relevantly.
- Many equations should be followed by either commas or full stops.
- In Eqs. 1, 2 & 3 I recommend to use normal font for “for” instead of italic.
- I suggest to give a rationale behind the exact form of Eqs. 7 & 14, especially to comment, why such specific numbers are used there.
- Line 273, page 8: there is something missing after “where:”.
- Lines 280-282, page 9: A paragraph-opening sentence is difficult to comprehend.
- Line 304, page 9: enphasizes -> emphasizes (typo)
- Line 370, page 12: dots over “i” are missing in some indices.
- Line 428, page 16: volatile -> volatility
- Eq. 20: Is there any argument why the authors use moduli in this equation instead of individual summed terms in Eq. 5?
- Eq. 20: The quantity H^OLHC_dk seems to be positive always (as Eq. 6 indicates), thus there is no need to use a modulus.
- Line 473, page 18: imbedded -> embedded (typo)
- Line 485, page 18: based historical -> based on historical
- Line 546, page 20: “For example, indeed,” seems not to correspond well with what follows in line 547.
- Figure 12: meaning of the arrows should be explained in caption.
Author Response
Thank you for your comments and suggestions! Very much appreciated, indeed!

Reviewer 3 Report
An interesting empirical research with a goal to propose a new indicator for market volatility. I think the authors did a good job providing variety of details regarding the proposed CSIE indicator. Since I have no major issues, I would suggest accepting the manuscript. Though I have few minor suggestions, which could be addressed prior to publication of the final paper.
Minor issues:
- Something seems to be wrong around line 273 (end of page 8 / start of page 9).
- I do not think calculating coefficient of variation makes sense for CSIE. Typically CV is calculated only for measures which lie on ratio scale. CSIE doesn't seem lie on ratio scale. It doesn't make sense to divide two estimates of CSIE, at least if the sign is kept. What is the difference between positive CSIE and negative CSIE values? What do they indicate?
- Differences between CSIE (4) and IE (15) should be discussed.
Author Response
Thank you for your comments and suggestions!
